# Wind Forces and Flow Patterns of Three Tandem Prisms with a Small Height–Width Ratio

**Kun Du [1],* and Bo Chen [2,3],***

[1] School of Civil Engineering, Beijing Jiaotong University, Beijing 100044, China
[2] School of Civil Engineering, Chongqing University, Chongqing 400044, China
[3] Chongqing Key Laboratory of Wind Engineering and Wind Energy Utilization, Chongqing 400044, China
* Correspondence: dukunbjtu@163.com (K.D.); chenbohrb@163.com (B.C.)

**Abstract:** Wind tunnel tests and large eddy simulations were conducted to investigate the dependency of wind forces and flow patterns on the spacing ($S$) for three tandem prisms with a small height–width ratio $H/W = 0.4$. At the spacing ratio $S/W = 0.7$, mean and root-mean-square drag of downstream prisms have large local peaks, and their magnitudes are larger than those at adjacent spacing ratios; these should be noted to ensure the safety and economy of the wind-resistant design of prism-like low-rise buildings. These phenomena are different from that of a small group of tandem prisms with a large $H/W$ and a large group of tandem prisms with a small $H/W$. At $S/W = 0.7$, tap pressure time histories of downstream prisms are non-stationary with abrupt changes, but wind force time histories of downstream prisms are stationary, unlike a small group of tandem prisms with a large $H/W$, where both tap pressure and win d force time histories are non-stationary. Above phenomena at $S/W = 0.7$ are attributed to a special asymmetric time-averaged wake regime, which has two modes with symmetric wake flow directions and they irregularly switch. The duration of each mode is ruleless. This special wake regime was not observed in previous studies on tandem prisms.

**Keywords:** wind interference; small height–width ratio prism; pressure coefficient; flow pattern; asymmetric wake

## 1. Introduction

Due to wind interference effects from surrounding prisms, wind forces and flow characteristics of a group of prisms is much different from those of an isolated prism. The flow patterns and interfered wind loads of a small group of three-dimensional (3D) prisms with a large height–width ratio ($H/W$) and two-dimensional (2D) prisms have been widely studied both experimentally [1–5] and numerically [6–8]. In contrast, for a small group of prisms with a small $H/W$, which is usually applied for factories and rural low-rise houses, few previous studies focus on the relations between its flow patterns among prisms and the wind loads acting on prisms have been conducted.

When a prism is placed in line with another prism parallel to the incident flow, the arrangement is called tandem arrangement. The flow patterns among the tandem 3D prisms with a large $H/W$ and tandem 2D square prisms were divided into four regimes depending upon the prism spacing [2,3], namely the reattachment flow ($S/W \leq 2$), bistable flow ($2 < S/W \leq 3.5$), stable synchronized flow ($3.5 < S/W \leq 50$), and unstable synchronized flow ($S/W > 50$), where $S$ is the clear spacing between two prisms and $W$ is the width of the windward wall of prisms. When the flow pattern of prisms changes from the reattachment flow to the bistable flow, the root-mean-square (RMS) fluctuating wind loads acting on 3D prisms with a large $H/W$ and 2D square prisms rapidly increased, and two types of instantaneous wake modes were observed in the prism gap. One mode is symmetric along the wind direction, while the other mode is asymmetric along the wind

direction; the time-averaged wake in the prism gap and mean pressures of the downstream prism are symmetric along the incident flow [9], which is ascribed to the fact that the instantaneous asymmetric flow in the prism gap is alternately and periodically biased to one of the side walls of the downstream prism. Sharm et al. [7] showed the flow field among tandem prism with $H/W = 7$ in the central vertical section at $0.5 < S/W \leq 9$. The results demonstrated that the upwash flow formed at the bottom behind the upstream prism had great effects on the flow between prisms.

For a small group of prisms with a small $H/W$, the interfered wind loads acting on prisms were usually studied by the pressure measurement with wind tunnel tests. Chand et al. [10] discussed mean wind pressure coefficients on a small group of prisms with $H/W = 0.7$. The results demonstrated that the staggered arrangement increased the suction on the roof of the downstream prism at the small prism spacing. Chen et al. [11] investigated the interference effects on the three prisms with a gable roof, where the $H/W$ was about 0.13, and there was a notable increase in the magnitude of negative mean pressure coefficients on the prisms at the small prism spacing. In addition, the flow patterns among a large group of prisms with $H/W < 1$ were divided into three categories depending on the prism spacing [12,13] including skimming flow, wake interference flow, and isolated roughness flow. However, few previous studies focused on the flow patterns of a small group of prisms with a small $H/W$, and the prism number had been confirmed as a key factor determining the wind interference effects of prisms [14]. Therefore, it is important to study the mechanism of flow patterns among a small group of prisms with a small $H/W$ and features of wind loads acting on prisms in different flow patterns.

In summary, to ensure both the safety and economy of the wind-resistant design of prism-like low-rise buildings, more studies should be conducted to investigate the wind loads and the mechanism of flow patterns of a small group of prisms with a small height–width ratio, as wind loads might be larger than those of a large group of prisms. This research performs wind tunnel tests and numerical simulations to examine the characteristics of wind loads and flow patterns of three tandem rectangular prisms with a small height–width ratio $H/W = 0.4$. The mean and root-mean-square (RMS) wind forces and time histories of pressures for prisms in different flow patterns are illustrated, and the mechanism of flow patterns is revealed by the flow fields around prisms.

## 2. Outline of Wind Tunnel Tests and Numerical Simulations

### 2.1. Pressure Measurements by Wind Tunnel Tests

Wind pressure measurements of three rectangular prisms with a small height–width ratio were carried out in a close circuit boundary layer wind tunnel located at Beijing Jiaotong University, China. The test section of the wind tunnel was 3.0 m wide and 2.0 m high. The spire-roughness technique was employed to simulate the atmospheric boundary layer over an open country terrain with a length scale of 1/200. As shown in Figure 1a, the measured longitudinal turbulence intensity profile ($I_u$) and mean wind velocity profile ($U/U_H$) with a power-law exponent $\alpha = 0.12$ followed the atmospheric boundary layer of the open terrain in the standard "Load Code for the Design of Building Structures (GB50009)" [15]. $U_H$ and $I_{u, H}$ are the mean wind speed and turbulence intensity at prism height $H$, where $U_H$ is 7.6m/s and $I_{u, H}$ is 0.11. The non-dimensional power spectrum densities of longitudinal velocity at $H$ is presented in Figure 1b. The corresponding turbulence integral scale in longitudinal wind direction is $L_u = 0.35$ m, which satisfies the similarity law for the length scale. The Reynolds number using the prism width $W$ and the mean velocity 7.6 m/s is $1.0 \times 10^5$.

The experimental prism arrangement and the wind direction of incident flow are displayed in Figure 2a, where three rectangular prisms are arranged along the long edge of the prism. The three prisms were the same size, which was 0.3 m long ($L$), 0.2 m wide, ($W$) and 0.08 m high ($H$), and the frontal $H/W$ of the prism was equal to 0.4. The geometric scale ratio was 1/200. In order to investigate interference effects of the prism spacing on

wind loads of three tandem prisms, sixteen spacing ratios of the clear spacing (*S*) to the width (*W*) of prisms *S/W* = 0.2, 0.4, 0.5, 0.6, 0.7, 0.8, 1, 1.2, 1.4, 1.6, 2, 2.4, 2.8, 3.2, 4, and 4.8 were considered. The definition of this tandem arrangement is the same as that in Sakamoto and Haniu [2]. The test on an isolated prism was also carried out. As shown in Figure 2b, 125 pressure taps were arranged on the roof and the four walls of each prism.

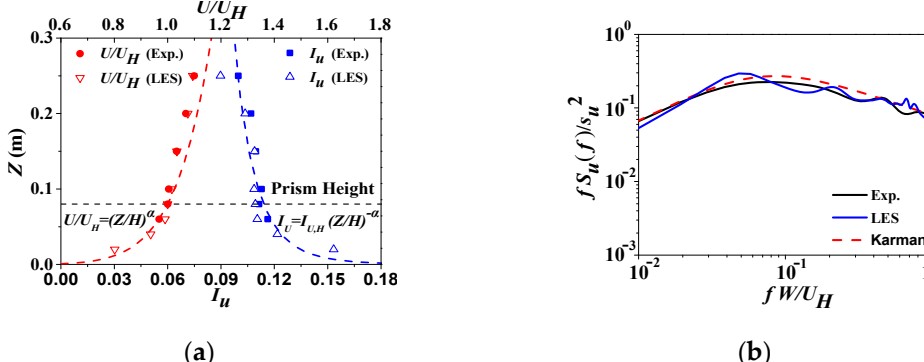

(**a**)                  (**b**)

**Figure 1.** Information of incident flow: (**a**) Simulated atmosphere boundary layer; (**b**) Longitudinal power spectrum density.

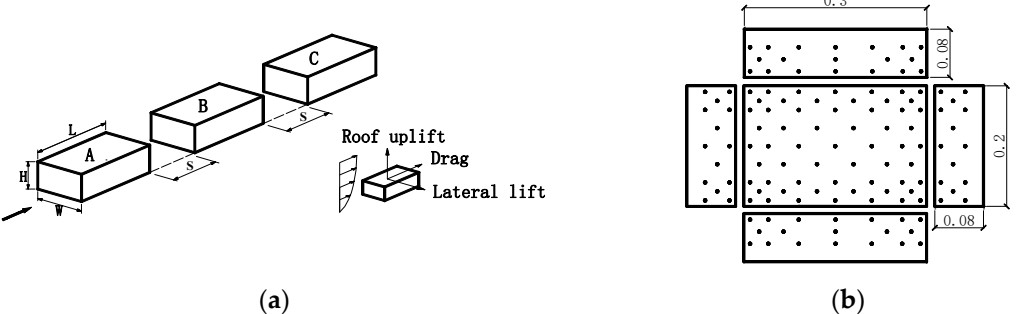

(**a**)                  (**b**)

**Figure 2.** Prism configuration and pressure tap arrangement: (**a**) Prisms arrangement and wind direction; (**b**) Prism pressure taps.

Fluctuating pressures on all three rectangular prisms were measured at the sampling frequency of 312 Hz. The mean wind speed at the top of the prism was 7.6 m/s, and the velocity scale ratio was 1/4. The sampling duration was about 90 s (about 75 min in reality). All measured fluctuating pressures were corrected with the frequency transfer function of the tube system [16]. The maximum blockage ratio was 0.2%, and no model blockage correction was conducted.

Fluctuating pressure coefficients $C_{pi}(t)$ of three rectangular prisms are defined as:

$$C_{pi}(t) = (P_i(t) - P_{static})/0.5\rho U_H^2 \tag{1}$$

where $P_i(t)$ is the measured pressure of tap *i* at time *t*; $\rho$ is the air density; and $U_H$ and $P_{static}$ are the free-stream velocity and the static pressure at the reference height, respectively, where the model does not affect the flow. The reference height is taken as the height of the prism *H*.

Investigated wind force coefficients include drag coefficients $C_D(t)$ (along-wind), lateral lift coefficients $C_{LL}(t)$ (cross-wind) in the horizontal direction, and roof uplift coefficients $C_{RL}(t)$ in the vertical direction, and they are defined as:

$$C_D(t) = F_D(t)/0.5\rho U_H^2 A_D \tag{2}$$

$$C_{LL}(t) = F_{LL}(t)/0.5\rho U_H^2 A_{LL} \tag{3}$$

$$C_{RL}(t) = F_{RL}(t)/0.5\rho U_H^2 A_{RL} \tag{4}$$

where $F_D$, $F_{LL}$, and $F_{RL}$ are the drag, the lateral lift and the roof uplift, respectively, as shown in Figure 2a. $A_D$, $A_{LL}$, and $A_{RL}$ are the projected area perpendicular to the direction of the three wind force components, respectively.

### 2.2. Large Eddy Simulations Setup

The CFD numerical simulation is adopted to analyze the wake characteristics among three prisms with the Ansys Fluent software package. Considering that the RANS models are not suitable to capture the unsteady wake among prisms, the 3D Large Eddy Simulations (LES) turbulence model is employed in this present study. The LES uses the filter function to separate the large and small eddies, and the large eddies directly calculate by the Navier–Stokes (N–S) equations, while the small eddies are calculated using a sub-grid scale (SGS) model. For the SGS model, the dynamic Smagorinsky–Lilly (DSL) is adopted. The pressure–velocity coupling uses the PISO algorithm. The second-order implicit scheme is selected for the temporal discretization, and the bounded central differencing is applied in momentum equations. The computational domain and boundary conditions are shown in Figure 3. The prism size is the same as the wind tunnel tests, and the blockage ratio is 1.5%, which is much smaller than the CFD best practice guidelines [17].

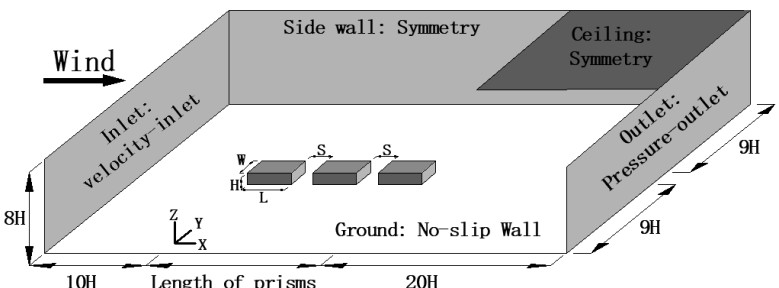

**Figure 3.** The schematic of computational domain and boundary conditions.

The fluctuating algorithm vortex method (VM) is used for fluctuating wind velocity inlet, which has been confirmed to predict the wind flow around the prism [18]. In this present study, the number of vortices is 200 [19]. At the inlet of the LES ($X/H = -10$), the wind flow with a turbulence intensity profile and a mean wind velocity profile are imposed. The incident flow gradually stabilizes after a distance development of *10H*, and then the simulated atmospheric boundary layer approaching the upstream prism ($X/H = 0$) is similar to that in the wind tunnel test, as shown in Figure 1a. The mean wind velocity and the turbulence intensity at the prism roof height are 7.6 m/s and 0.11, respectively. The simulated longitudinal turbulence spectrum at the prism roof height is presented in Figure 1b, and the corresponding turbulence integral scale in longitudinal wind direction is $L_u$ = 0.28 m. The Reynolds number using the prism width and the mean velocity 7.6 m/s is about $1.0 \times 10^5$ in LES, which is consistent with wind tunnel tests.

To check the grid sensitivity, results based on three types of grids (coarse, basic, and fine) in the case of *S/W* = 0.7 are compared. The numbers of computational meshes for coarse grids, basic grids, and fine grids are $6.6 \times 10^6$, $1.0 \times 10^7$, and $1.4 \times 10^7$, respectively. The average value of the non-dimensional wall distance $y^+$ for the three types of grids is close to 1, and the average values of non-dimensional wall distances $x^+$ and $z^+$ for the three types of grids are both close to 4. The computational mesh details of basic grids around three prisms are shown in Figure 4, and the coarse grids and fine grids are not shown for brevity. The non-dimensional time steps $\Delta t^* = \Delta t U_H/H$ in terms of coarse

grids, basic grids, and fine grids are 0.04, 0.02, and 0.01, respectively. For all simulations, the maximum courant numbers are less than 1.5. To ensure the stable turbulent condition in the computation domain, the data are recorded after 2 s from the beginning of the simulation, and the sampling frequency is 312 Hz, which is the same as wind tunnel tests. Because the LES transient calculation consumes much time and computational cost, all simulations are sampled for 12 s (about 10min in reality), except for the special case of *S*/*W* = 0.7, which is sampled for 30 s (about 25 min in reality).

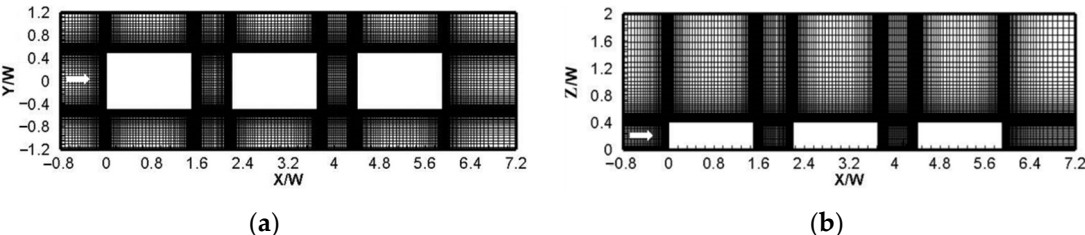

**(a)** **(b)**

**Figure 4.** Computational meshes near the prisms model (Basic grids): (**a**) Horizontal section at *Z*/*H* = 0.5; (**b**) Vertical section at *Y*/*W* = 0.

Figure 5 shows the relative difference $\varepsilon$ ($\varepsilon = |C_{EXP} - C_{Numerical}|/C_{EXP}$) for mean and RMS forces coefficients of three prisms (*S*/*W* = 0.7) between the results of the wind tunnel tests and the results of the numerical simulations using three types of grids. As shown in Figure 5, the difference between the mean and RMS coefficients for $C_D$ and $C_{RL}$ using the coarse grids and those using the basic grids is large, whereas the result difference between the fine grids and basic grids is small. The maximum values of the relative difference $\varepsilon$ for mean and RMS forces coefficients of numerical simulations using basic grids are 14.3% and 13.9%, respectively, where the deviations are less than 20% and can be acceptable [20]. All these above forces coefficients are calculated using the first 12 s sampling duration, including the results of wind tunnel tests. The above results indicate that the computational mesh size of basic grids can obtain good computational accuracy. Therefore, the configuration of basic grids is employed for other simulation cases with different prism spacing ratios, and the height ratio is 1.1. The minimum size of computational meshes is refined up to 0.04 mm on the no-slip wall of the ground and the model.

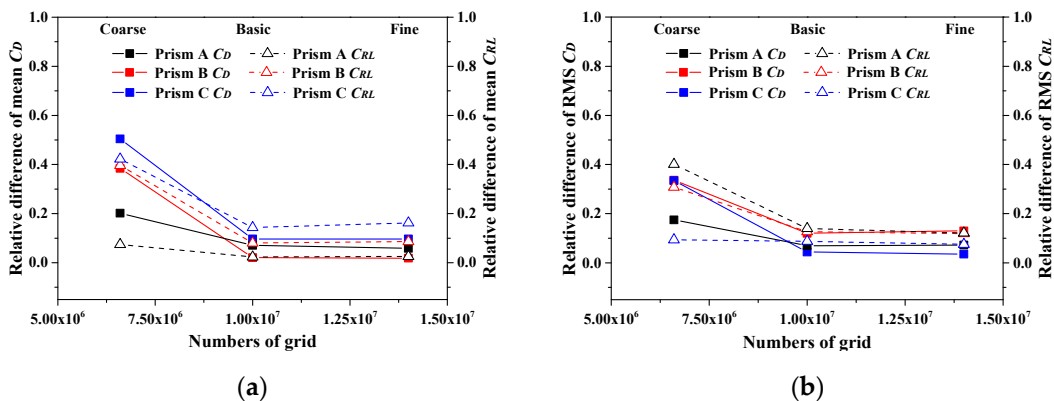

**(a)** **(b)**

**Figure 5.** Relative difference ($\varepsilon$) of wind forces between the experimental and LES: (**a**) Mean $C_D$ and Mean $C_{RL}$; (**b**) RMS $C_D$ and RMS $C_{RL}$.

## 3. Results and Discussions

### 3.1. Wind Pressure Coefficients on the Isolated Prism

Figure 6 displays the mean and RMS pressure coefficient distributions on the isolated prism obtained by wind tunnel tests, when the wind direction of incident flows along the long edge of the prism. Large magnitudes of negative mean pressure coefficients and RMS

pressure coefficients are observed near the leading roof edge and wall edges due to the flow separation. The magnitudes of mean pressure coefficients and RMS pressure coefficients decrease rapidly along the wind direction from the separation area, and the rate of change gradient of mean pressure coefficients is much larger than that of RMS pressure coefficients. The distributions and magnitudes of roof mean pressure coefficients are similar to the results of Stathopoulos and Dumitrescu [21]. Table 1 shows the mean and RMS force coefficients, which are calculated using the 90s duration pressure time histories in wind tunnel tests.

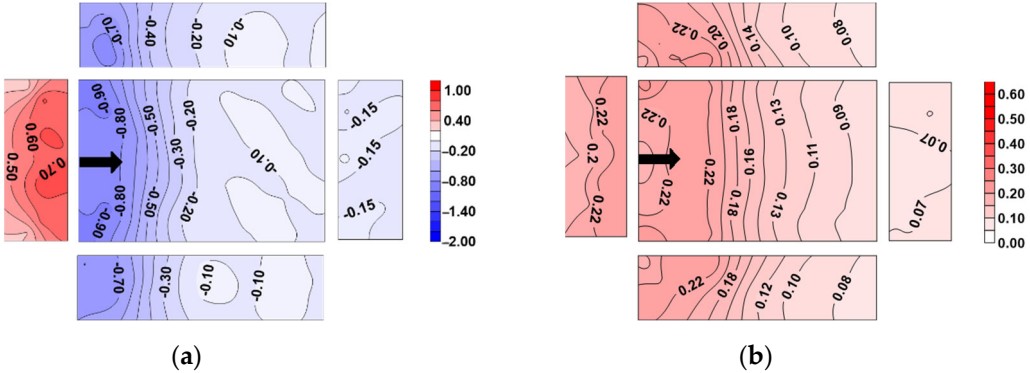

**Figure 6.** Pressure coefficients of the isolate prism: (**a**) Mean pressure coefficients; (**b**) RMS pressure coefficients.

**Table 1.** Wind force coefficients of the isolated prism.

| Mean $C_D$ | RMS $C_D$ | Mean $C_{LL}$ | RMS $C_{LL}$ | Mean $C_{RL}$ | RMS $C_{RL}$ |
|---|---|---|---|---|---|
| 0.73 | 0.14 | 0.01 | 0.07 | 0.35 | 0.06 |

*3.2. Dependency of Wind Force Coefficients on Prism Spacing*

This section analyzes the interference effects of the clear spacing on the wind force coefficients of the three tandem prisms. The mean and RMS coefficients of wind forces were calculated from the 90 s duration pressure time histories in wind tunnel tests. To quantitatively evaluate interference effects on wind forces, interference factors (*IF*) of drag and roof uplift coefficients are defined as Equations (5) and (6), respectively. Since the magnitude of the lateral lift coefficient of the isolated prism is close to zero, the lateral lift coefficients of prisms are not expressed in the form of interference factors to avoid being divided by zero.

$$IF_{C_D} = C_{D,disturbed}/C_{D,isolated} \tag{5}$$

$$IF_{C_{RL}} = C_{RL,disturbed}/C_{RL,isolated} \tag{6}$$

where the numerator and denominator are the wind force coefficients of the disturbed prism and isolated prism, respectively.

Figure 7 shows the variation of coefficients of mean wind forces and RMS wind forces of three tandem prisms with the spacing ratio (*S/W*). As shown in Figure 7, the wind force coefficients of tandem prisms with *S/W* in previous studies [2,13,22] are compared with those in this present study. The displayed previous results include mean and RMS drag coefficients of the downstream prism with *H/W* = 3 in Sakamoto and Haniu [2], mean and RMS drag coefficients of the downstream 2D square prism in Lu et al. [22], and coefficients of mean drag and mean roof uplift of the downstream prism with *H/W* = 0.5 immersed in a large group of prisms in Hussain and Lee [13], where the RMS wind force coefficients were not discussed.

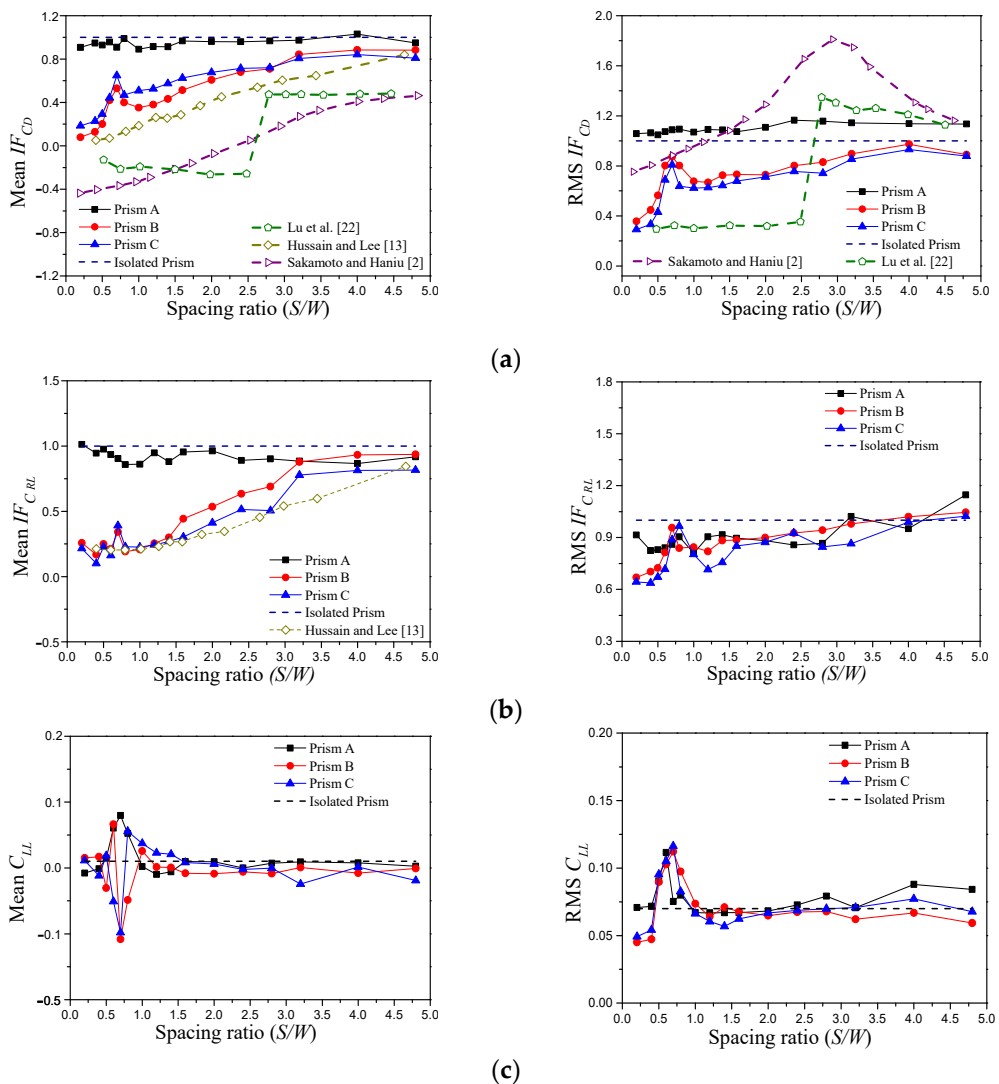

**Figure 7.** Interfered wind forces of three tandem prisms: (**a**) Interference factors of drag coefficients; (**b**) Interference factors of roof uplift coefficients; (**c**) Lateral lift coefficients.

It can be found that the mean and RMS coefficients of the drag and roof uplift of the upstream prism A are not sensitive to the variation of the spacing, and their values are close to those of the isolated prism.

As shown in Figure 7, when spacing ratios are $S/W \leq 0.4$, shielding effects among prisms are significant, which makes interference factors of mean drag coefficients of downstream prisms B and C very small, and these are similar to results in [13]. According to the flow patterns demonstrated in Hunter et al. [23], this flow pattern among prisms at $S/W \leq 0.4$ is regarded as the skimming flow regime. In addition, for the two tandem prisms with $H/W = 3$ [2], the negative mean drag coefficients of the downstream prism were observed at small spacing ratios (Figure 7a). This phenomenon is not found among the tandem prisms with $H/W = 0.4$ in this present study. At spacing ratios $S/W \leq 0.4$, the other wind force coefficients of downstream prisms except for the mean lateral lift are small compared with those of the isolated prism.

It is interesting to note from Figure 7 that when the spacing ratios are $0.4 < S/W \leq 0.8$, the mean and RMS interference factors of drag and lateral lift of the downstream prisms B and C have large local peaks, increase and decrease rapidly in this prism spacing range, and then increase with the increase of the prism spacing at $S/W > 0.8$. The RMS $IF_{C_{RL}}$ of downstream prisms and mean $IF_{C_{LL}}$ and RMS $IF_{C_{LL}}$ of prism A also have similar

variations at $0.4 < S/W \leq 0.8$, and the mean $IF_{C_{RL}}$ reaches the local maximum at $S/W = 0.7$. As shown in Figure 7a, for two tandem prisms with $H/W = 3$ in Sakamoto and Haniu [2], a similar trend of RMS drag of the downstream prism was observed at $2 < S/W \leq 3.5$, and there was a local peak at $S/W = 3$, but the mean drag of the downstream prism increased monotonically with the increase of $S/W$, and there was no local peak at $2 < S/W \leq 3.5$. The coefficients of mean and RMS drag of the downstream 2D square prism rapidly increased at $S/W = 2.7$ and always decreased slowly with the increase of the prism spacing at $S/W > 2.7$ (Figure 7a). These results of 2D rectangular prisms arranged in tandem were also reported in Zhang et al. [24], and the aspect ratio ($L/W$) of the prism only affects magnitudes of wind forces but does not affect their variation trends with $S/W$. Furthermore, for a large group of prisms with $H/W = 0.5$ in Hussain and Lee [13], the mean drag and mean roof uplift of the downstream prism gradually increased with the increase of $S/W$ without local peaks, as shown in Figure 7a,b.

When spacing ratios range from 0.8 to 2.8, mean and RMS coefficients of drag and roof uplift of downstream prisms B and C continuously increase with the increase of $S/W$. The flow pattern among prisms tends to be the wake interference flow regime [23].

As shown in Figure 7, when $S/W > 2.8$, mean and RMS coefficients of drag and roof uplift of downstream prisms increase slightly with the increase of $S/W$, and interference effects among prisms become weak. The downwind flow of the upstream prism travels a sufficient distance before encountering the downstream prism, and the flow pattern tends to be the isolated roughness flow [23]. The values of mean and RMS wind forces coefficients of three prisms are shown in Tables A1–A3 in Appendix A.

### 3.3. Wind Pressure Time Histories on Three Tandem Prisms

Characteristics of wind pressure time histories of the prism are influenced by the different flow patterns around prisms. At present, few studies discuss the characteristics of pressure time histories on a group of tandem prisms with a small $H/W$ for different flow regimes, and they will be discussed in this section. The pressure coefficient time histories of twelve pressure taps on the windward wall and leeward wall of three prisms are displayed at three typical spacing ratios (0.2, 0.7, and 1.2) in Figure 8. To reflect the fluctuating characteristics of pressure signals of twelve taps clearly, only a partial sampling duration from 30s to 80s is displayed, as shown in Figure 8.

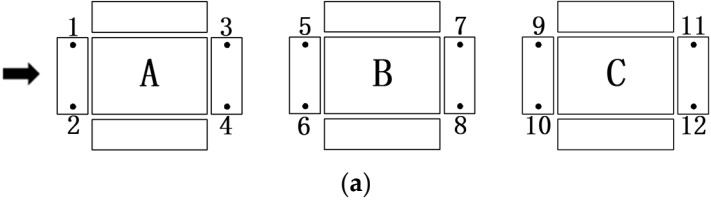

(**a**)

(**b**)

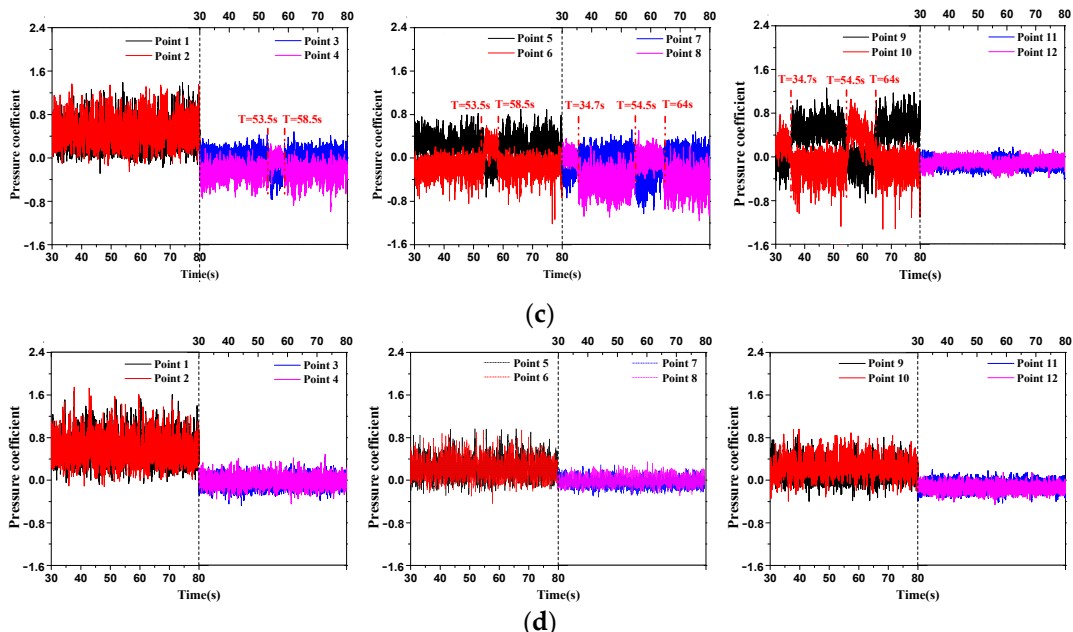

**Figure 8.** Pressure coefficient time histories in taps of prisms: (**a**) Pressure taps on windward wall and leeward wall; (**b**) Spacing ratio: 0.2; (**c**) Spacing ratio: 0.7; (**d**) Spacing ratio: 1.2.

It can be seen that the fluctuating characteristics of pressures on the windward wall of the upstream prism A are basically similar at different spacing ratios, but those of the downstream prisms B and C are significantly different at the three spacing ratios. As shown in Figure 8b, when the spacing ratio is $S/W = 0.2$, the pressure coefficients on the windward wall of downstream prisms are much smaller in magnitude than those on the upstream prism A due to shielding effects.

When the spacing ratio is $S/W = 0.7$, the pressure coefficient time histories in local pressure taps of downstream walls show abrupt jump or plunge, which is non-stationary during the sampling duration, as shown in Figure 8c. The fluctuating pressures of the leeward taps of prism A and the windward taps of prism B experience abrupt change at about $T = 53.5$ s, and they restore to the original state after about $T = 58.5$ s. As shown in Figure 8c, for the leeward wall of prism B and the windward wall of prism C, there are three abrupt changes in the fluctuating pressures at about $T = 34.7$ s, $T = 54.5$ s, and $T = 64$ s. Compared with the windward wall of prism B, the fluctuating pressures of the windward wall of prism C undergo multiple jumps and plunges. The time moments of irregular abrupt changes in tap pressure time histories within different prism gaps are not related.

When the spacing ratio is $S/W = 1.2$, the fluctuating pressures of downstream taps change from non-stationary to stationary, as shown in Figure 8d.

To analyze the influence of the abrupt changes in fluctuating pressures on downstream prisms at $S/W = 0.7$, Figure 9 shows the drag coefficient time histories of three prisms at $S/W = 0.7$. Different from the non-stationary fluctuating pressures on downstream prisms, the fluctuating drag coefficients of downstream prisms are stationary. As shown in Figure 7a, the mean and RMS drag coefficients of upstream prism A are always similar to those of the isolated prism, which reflects that the drag coefficient of prisms mainly depends on wind loads of the windward wall. Since wind loads of the windward wall of upstream prism A are not affected by the interference effect among prisms at the small spacing ratio, the magnitude of fluctuating drag coefficients of upstream prism A is larger than that of downstream prisms B and C, as shown in Figure 9. When the spacing ratio $S/W = 0.7$, the upstream prism A has significant shielding effects on downstream prisms B and C, and wind loads of the windward wall of downstream prisms are reduced, which results in the small magnitude of fluctuating drag coefficients

of downstream prisms B and C. In addition, the fluctuating drag coefficients of downstream prisms B and C are basically the same, and the reason is that the characteristics of the flow field between the downstream prisms are similar [25].

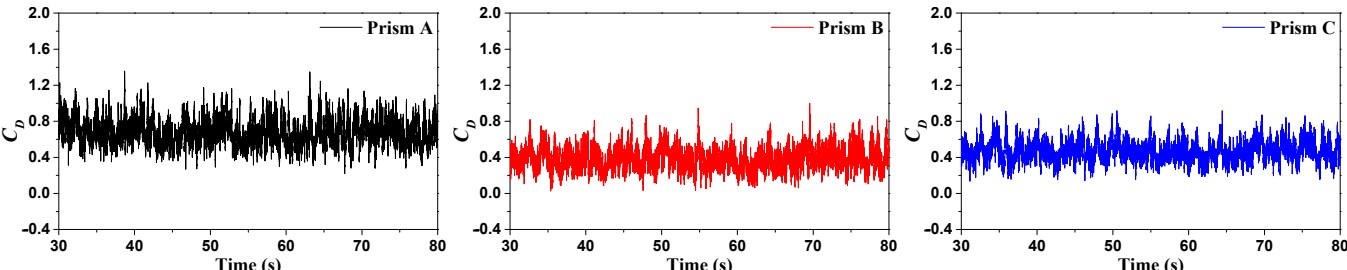

**Figure 9.** Drag coefficient time histories of three prisms at the spacing ratio of 0.7.

At the spacing ratio of 0.7, the non-stationary characteristics of tap pressures signals of the leeward wall of prism B and the windward wall of prism C are more obvious than those of the leeward wall of prism A and the windward wall of prism B. The mean and RMS pressure coefficient distributions on these walls during two sampling periods are analyzed to illustrate the influence of the abrupt changes in the fluctuating pressures on the mean and RMS pressure coefficients of downstream walls.

As shown in Figures 10a and 11a, period C-1 (42 s–45 s) and period C-2 (57 s–60 s) are extracted from the sampling duration (40 s–70 s). The pressure sign of point 9 is positive in period C-1, but it is negative in period C-2. As shown in Figures 10 and 11, the mean and RMS pressure coefficient distributions on the leeward wall of prism B and the windward wall of prism C are all asymmetric for these two sampling periods. These asymmetric pressure distributions on downstream prisms are not observed in previous studies for two tandem prisms with a large $H/W$, two tandem 2D prisms, and a large group of prisms with a small $H/W$. The characteristics of asymmetric distributions of the mean and RMS pressure coefficients in period C-1 are symmetric to those in period C-2. Hence, during the sampling duration, the asymmetric mean and RMS pressure distributions on downstream prisms are unstable at the spacing ratio $S/W = 0.7$ due to the non-stationary fluctuating pressures with irregular abrupt changes, and these pressure distributions are strongly sensitive to the number of abrupt changes during the sampling duration and the length of the sampling duration. In constrast, because the pressure distributions on the downstream prism before and after the abrupt change in fluctuating pressures are symmetric with each other (Figures 10 and 11), the overall wind loads on the downstream prism are little affected by the non-stationary fluctuating pressures on downstream prisms at $S/W = 0.7$. In addition, drag time histories of the downstream prism are stationary during the sampling duration, as shown in Figure 9. For the tandem 3D prisms and 2D square prisms at $2 < S/W \leq 3.5$ in the bistable flow regime [2,3], a symmetric instantaneous wake and an asymmetric instantaneous wake irregular appear in the prism gap, which will induce the tap pressure time histories and wind force time histories for the downstream prism to be non-stationary. These are significantly different from the stationary wind force time histories in this present study.

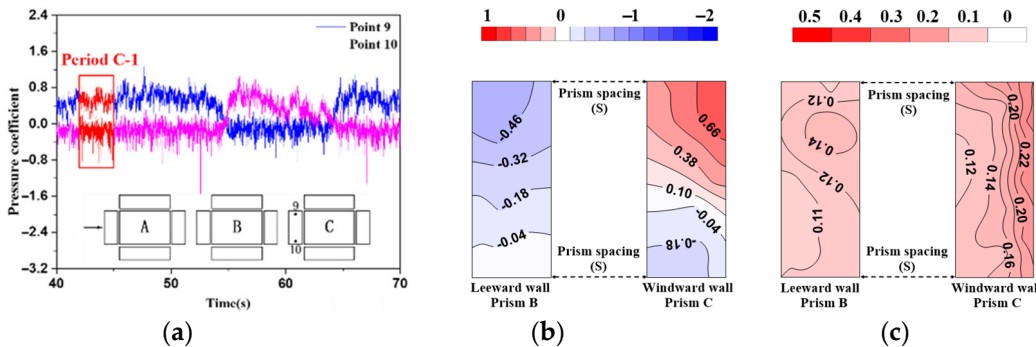

**Figure 10.** Pressure coefficients on walls in period C-1 (Spacing ratio: 0.7): (**a**) Period C-1 (42 s–45 s); (**b**) Mean $C_p$; (**c**) RMS $C_p$.

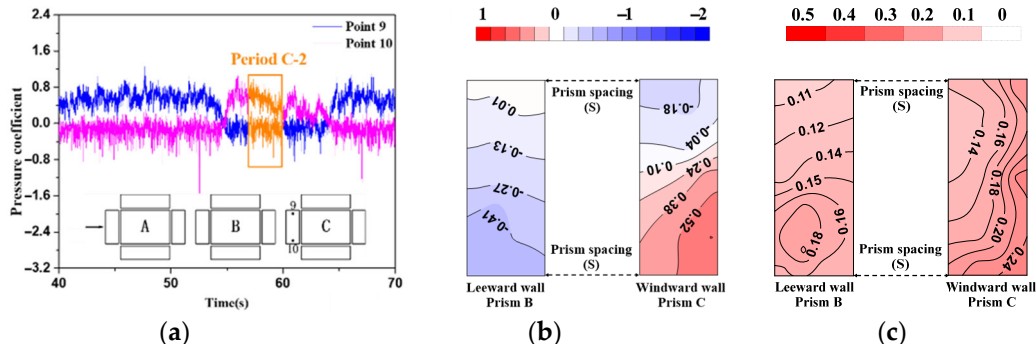

**Figure 11.** Pressure coefficients on walls in period C-2 (Spacing ratio: 0.7): (**a**) Period C-2 (57 s–60 s); (**b**) Mean $C_p$; (**c**) RMS $C_p$.

To verify and understand the abrupt changes in the pressure coefficient time histories of downstream prisms (Figure 8c) obtained from the wind tunnel tests, a total of 30s sampled time (about 25 min in reality) is performed by LES simulation at the spacing ratio of 0.7. Figure 12 shows the pressure coefficient time histories of taps at the same positions of wind tunnel tests in Figure 8a. As shown in Figure 12, the pressure coefficient time histories of points (3–6) and points (7–10) suddenly change at about T = 16.5 s and T = 23.5 s, respectively. This abrupt change in LES is similar to that in wind tunnel tests, but only one abrupt change occurs during the 30s sampling duration in numerical simulations.

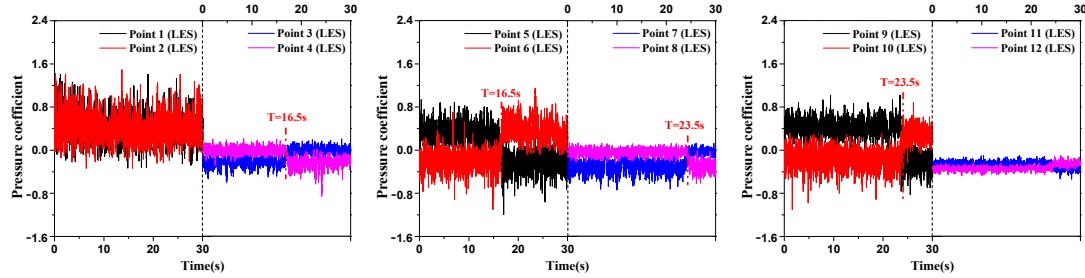

**Figure 12.** Simulated pressure coefficient time histories in taps of prisms (Spacing ratio: 0.7).

Figures 13 and 14 show the mean and RMS pressure coefficients on the leeward wall of prism B and the windward wall of prism C during the sampled period 9 s to 12 s and the sampled period 25 s to 28 s in LES. It can be seen that the asymmetric distributions and magnitudes of mean and RMS pressure coefficients in LES displayed reasonably good agreement with those in the wind tunnel tests. Compared with the wind tunnel tests, the

pressure distributions on the downstream prism before and after the abrupt change in fluctuating pressures are also symmetric with each other.

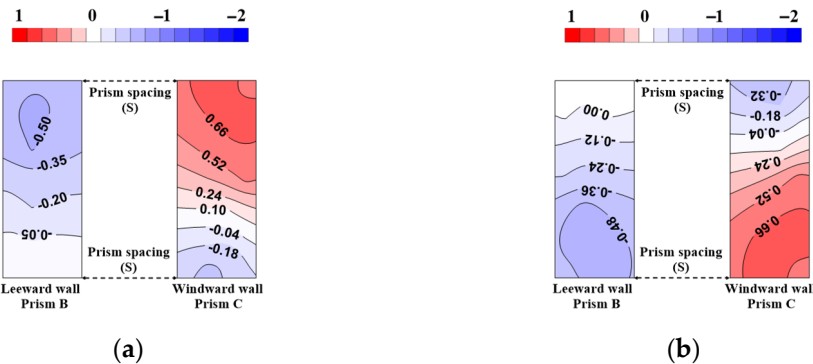

**Figure 13.** Mean pressure coefficient on walls (Spacing ratio: 0.7): (**a**) Sampled period (9 s to 12 s); (**b**) Sampled period (25 s to 28 s).

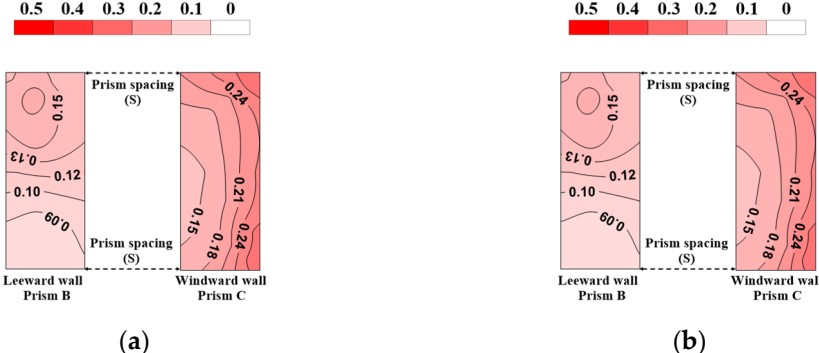

**Figure 14.** RMS pressure coefficient on walls (Spacing ratio: 0.7): (**a**) Sampled period (9 s to 12 s); (**b**) Sampled period (25 s to 28 s).

*3.4. Wind Flow among Three Tandem Prisms*

The wake characteristics among three tandem prisms at three typical spacing ratios of 0.2, 0.7, and 1.2 were simulated by LES. Figures 15 and 16 show the time-averaged velocity–magnitude contours and time-averaged streamlines of the central horizontal section (*Z* = 0.5*H*) and central vertical section (*Y* = 0) around the isolated prism and the group of prisms. The statistical duration for case *S/W* = 0.2 and case *S/W* = 1.2 is from 0 s to 12 s (about 10 min in reality), and there are two statistical durations for case *S/W* = 0.7, which are 0 s to 12 s and 24 s to 30 s. As shown in Figure 12, for the statistical duration from 0 s to 12 s, there is no abrupt change in pressure time histories in local pressure taps of downstream prisms, and for the statistical duration from 24 s to 30 s, the pressure time histories in local pressure taps of downstream prisms B and C both have undergone abrupt changes.

As shown in Figure 15a, the incoming flow separates at the leading edge of prism side walls and forms a pair of symmetrical separated vortices around the isolated prism. A pair of symmetrical wake vortices are also observed behind the prism, which was satisfied with the symmetric time-averaged wake behind the isolated rectangular prism reported in previous studies [19,26]. It can be seen from Figure 16a that in the central vertical section of the isolated prism, three vortices are formed and respectively appear at the windward wall bottom, leading roof edge, and leeward wall bottom.

When the spacing ratios *S/W* ≤ 0.4, the flow pattern around three prisms belongs to the skimming flow regime. The flow above the upstream prism roof skims over the prism gaps, and separation bubbles are not observed at the leading roof edge of downstream prisms (Figures 15b and 16b). Meanwhile, two small vortices appear in the prism gap near

the two edges of the leeward wall of the upstream prism, respectively, as shown in Figure 15b.

When the flow pattern transits from skimming flow to wake interference flow at spacing ratios $0.4 < S/W \leq 0.8$, shielding effects of the upstream prism gradually decrease, and the prism gap flow behind the upstream prism accelerates and strongly rolls up into the prism gaps, as shown in Figure 15c-1,c-2. Before the abrupt changes in pressure time histories of downstream prisms, the simulated time-averaged streamlines are asymmetric between prisms. A single recirculating zone at one side of the leeward wall of prisms A and B is observed in Figure 15c-1. Such asymmetric flow in the prism gap produces the asymmetric mean and RMS pressure distributions on downstream prisms, as shown in Figures 13 and 14. When the pressure time histories in local pressure taps of downstream prisms have undergone abrupt changes, the direction of wake flow between the prisms in Figure 15c-1 changes into the other side of prisms in Figure 15c-2, and these two directions of wake flow are basically symmetric along the direction of the incident flow. The flow field among prisms in the central vertical section is the same in Figure 16c-1,c-2. These flow characteristics illustrate that the non-stationary fluctuating pressures on downstream prisms in Figure 12 are induced by the rapid alteration in the bias direction of asymmetric flow, and there are two types of asymmetric wake modes with different wake flow directions between prisms. The switching interval between the two asymmetric wake modes is irregular, which causes the moment of the abrupt change in pressure time histories in local pressure taps of downstream prisms to be ruleless. However, this symmetrical switching between the two asymmetric wake modes (Figure 15c-1,c-2) does not affect stationarity of the drag time histories, lateral lift, and roof uplift of the downstream prisms.

This asymmetric time-averaged wake in this present study was not observed in previous studies on the flow field for two tandem prisms with a large *H/W*, two tandem 2D prisms, and a large group of prisms with a small *H/W* [7,9,23,27,28]. As shown in Figures 15a and 16a, the wake behind the isolated prism with a small *H/W* is mainly controlled by the side flow and roof flow. Compared with the prism with a large *H/W* [7,29], there is no upwash flow formed at the bottom behind the prism in this present study, where only one vortex is observed behind the prism in the central vertical section in Figure 16c-1,c-2. When the spacing ratio is *S/W* = 0.7, the wake vortices behind the upstream have incomplete development and are significantly different from those behind the isolated prism. Because the stable upwash flow is not formed behind the prism with a small *H/W*, the incomplete development wake vortices are easily affected by the unsteady turbulence flow near the ground and form asymmetric time-averaged flow in the prism gap. As the *S/W* increases, this asymmetric time-averaged wake will turn into the symmetric time-averaged wake, and the critical *S/W* of the transition is related to the along-wind dimension of complete development circulation vortices behind the upstream prism, where the unsteady turbulence flow near the ground has little effect on the complete development wake circulation vortices [30]. For the flat roof prism with a small *H/W*, this critical spacing is approximately 2*H* (0.8*W*) [20,31], which is equal to the critical *S/W*, where the mean and RMS wind forces of downstream prisms end rapid alteration. This illustrates that the rapid increase and decrease in the wind forces of downstream prisms at $0.4 < S/W \leq 0.8$ in Figure 7 are caused by the emergence and disappearance of asymmetric wake in the gap.

Further, as the prism spacing ratio increases to 1.2, the flow pattern around the prisms is the wake interference flow regime. The prism gap distance is sufficient to form two circulation vortices (Figure 15d), and the flow separation is observed at the leading roof edge of downstream prisms (Figure 16d).

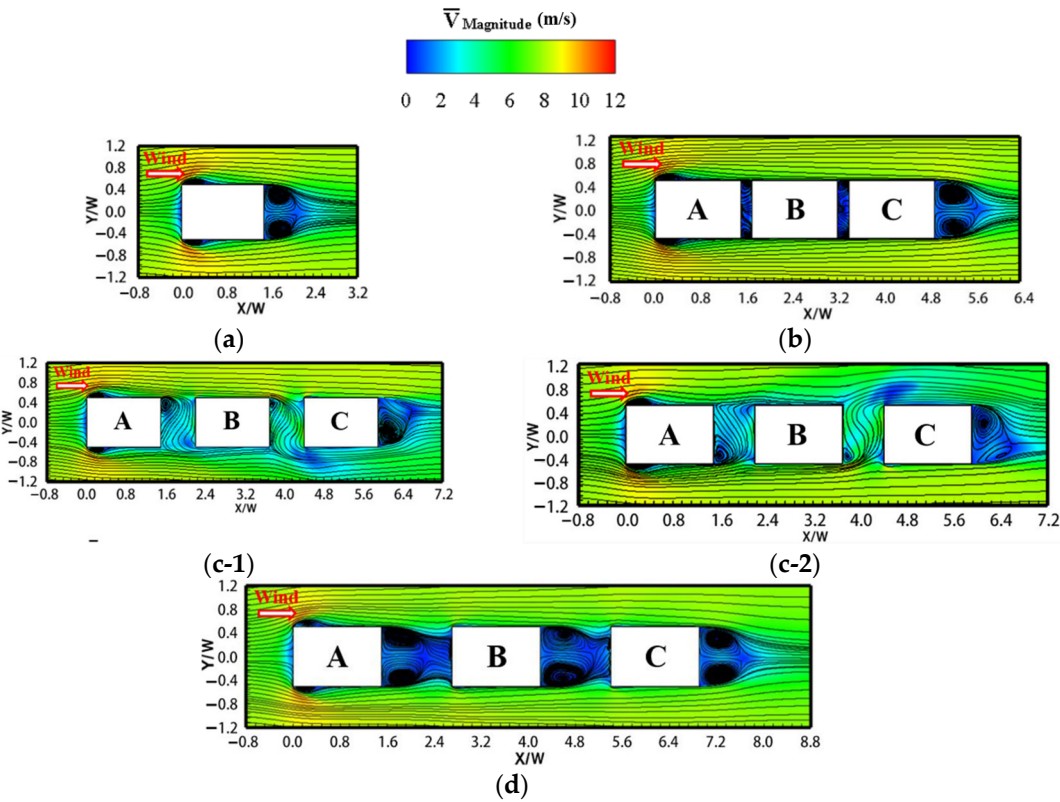

**Figure 15.** Velocity magnitude and streamlines of horizontal section (*Z* = 0.5*H*): (**a**) Isolated prism; (**b**) Spacing ratio: 0.2; (**c-1**) Spacing ratio: 0.7 (Period 0 s–12 s); (**c-2**) Spacing ratio: 0.7 (Period 24 s–30 s); (**d**) Spacing ratio: 1.2.

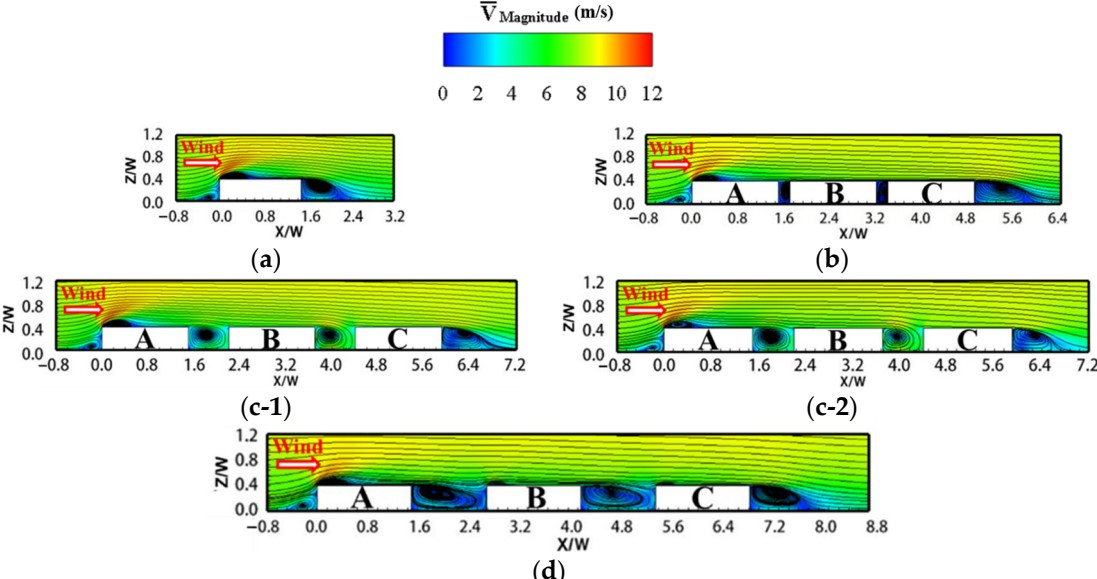

**Figure 16.** Velocity magnitude and streamlines of vertical section (*Y* = 0): (**a**) Isolated prism; (**b**) Spacing ratio: 0.2; (**c-1**) Spacing ratio: 0.7 (Period 0 s–12 s); (**c-2**) Spacing ratio: 0.7 (Period 24 s–30 s); (**d**) Spacing ratio: 1.2.

## 4. Conclusions

Wind tunnel tests and LES were conducted to investigate the dependency of wind loads and the mechanism of flow patterns on the clear spacing for three tandem rectangular prisms with a small height–width ratio ($H/W = 0.4$). The following conclusions were derived from this present study:

The coefficients of mean and RMS drag and RMS roof uplift of downstream prisms have large local peaks and increase rapidly and then decrease rapidly at the spacing ratio $0.4 < S/W \leq 0.8$, and the mean roof uplift coefficient reaches the local peak at $S/W = 0.7$. This phenomenon is much different from that of a small group of tandem prisms with a large $H/W$ and a large group of tandem prisms with a small $H/W$. In addition, the above local peaks of wind forces of prisms at $S/W = 0.7$ are much larger than wind forces of prisms at the adjacent spacing ratios. If the above phenomenon is not found in the experiment, the wind loads for the design of low-rise buildings at $S/W = 0.7$ will be significantly underestimated, resulting in potential safety hazards. Furthermore, at $S/W = 0.7$, the tap pressure time histories of downstream prisms are non-stationary with abrupt changes at irregular moments. Within the tap pressure time histories without abrupt changes, mean and RMS pressure distributions on downstream prisms are asymmetric, but these are symmetric to those after the abrupt changes in pressure time histories. Compared with the non-stationary tap pressure time histories, wind forces time histories of the downstream prism are stationary at $S/W = 0.7$, and these are much different from those for a small group of tandem prisms with a large $H/W$, where both tap pressure and wind force time histories of the downstream prism are non-stationary.

The mechanism of the flow pattern at $S/W = 0.7$ was analyzed by LES, and the asymmetric time-averaged wake regime was observed, which reveals that the asymmetric pressure distributions on downstream prisms are attributed to the asymmetric flow between prisms. There are two types of asymmetric wake modes between prisms. The duration of each asymmetric wake mode is ruleless, and the irregular switching between two modes in the prism gap causes the abrupt changes in the pressure time histories of downstream prisms. Because the bias directions of two asymmetric flow are symmetric along the incident flow, the irregular switching in wake modes has little effect on the wind force pressure time histories of downstream prisms and keeps these wind forces stationary. In addition, the rapid increase and decrease in the wind forces of downstream prisms at $0.4 < S/W \leq 0.8$ are caused by the emergence and disappearance of the asymmetric wake between prisms. This special asymmetric time-averaged wake regime in this paper was not observed in previous studies on flow fields around a group of tandem prisms.

It should be mentioned that the geometric characteristics of prisms are one key factor to affect flow patterns among prisms, and the study of this topic will be carried out in the future.

**Author Contributions:** All authors contributed significantly to this study. K.D. collected the data, performed the analysis, and wrote the paper; B.C. revised the paper. All authors have read and agreed to the published version of the manuscript.

**Funding:** This research was funded by the Fundamental scientific research business expenses of Beijing Jiaotong University, grant numbers C17JB00250.

**Institutional Review Board Statement:** Not applicable.

**Informed Consent Statement:** Not applicable.

**Data Availability Statement:** Not applicable.

**Acknowledgments:** The authors are grateful to Lina Zhang from Beijing Jiaotong University for her contribution to the paper.

**Conflicts of Interest:** The authors declare no conflict of interest.

## Appendix A

**Table A1.** Mean and RMS wind force coefficients of prism A.

| Wind Force | Spacing Ratio (*S*/*W*) | | | | | | | | | | | | | | | |
|---|---|---|---|---|---|---|---|---|---|---|---|---|---|---|---|---|
| | 0.2 | 0.4 | 0.5 | 0.6 | 0.7 | 0.8 | 1 | 1.2 | 1.4 | 1.6 | 2 | 2.4 | 2.8 | 3.2 | 4 | 4.8 |
| Mean $C_D$ | 0.662 | 0.692 | 0.678 | 0.698 | 0.664 | 0.721 | 0.650 | 0.668 | 0.667 | 0.705 | 0.701 | 0.700 | 0.705 | 0.710 | 0.752 | 0.693 |
| Mean $C_{RL}$ | 0.354 | 0.331 | 0.342 | 0.327 | 0.317 | 0.300 | 0.301 | 0.332 | 0.308 | 0.334 | 0.337 | 0.312 | 0.316 | 0.310 | 0.303 | 0.321 |
| Mean $C_{LL}$ | 0.008 | 0.001 | 0.015 | 0.060 | 0.080 | 0.052 | 0.002 | 0.010 | 0.006 | 0.010 | 0.009 | 0.000 | 0.007 | 0.009 | 0.008 | 0.003 |
| RMS $C_D$ | 0.148 | 0.149 | 0.147 | 0.150 | 0.152 | 0.153 | 0.150 | 0.153 | 0.152 | 0.150 | 0.155 | 0.163 | 0.162 | 0.160 | 0.159 | 0.159 |
| RMS $C_{RL}$ | 0.055 | 0.049 | 0.050 | 0.050 | 0.052 | 0.054 | 0.049 | 0.054 | 0.055 | 0.054 | 0.053 | 0.051 | 0.052 | 0.061 | 0.057 | 0.069 |
| RMS $C_{LL}$ | 0.071 | 0.072 | 0.090 | 0.112 | 0.075 | 0.080 | 0.067 | 0.067 | 0.067 | 0.067 | 0.068 | 0.073 | 0.079 | 0.071 | 0.088 | 0.084 |

**Table A2.** Mean and RMS wind forces of prism B.

| Wind Force | Spacing Ratio (*S*/*W*) | | | | | | | | | | | | | | | |
|---|---|---|---|---|---|---|---|---|---|---|---|---|---|---|---|---|
| | 0.2 | 0.4 | 0.5 | 0.6 | 0.7 | 0.8 | 1 | 1.2 | 1.4 | 1.6 | 2 | 2.4 | 2.8 | 3.2 | 4 | 4.8 |
| Mean $C_D$ | 0.057 | 0.094 | 0.146 | 0.307 | 0.386 | 0.292 | 0.257 | 0.278 | 0.315 | 0.374 | 0.444 | 0.496 | 0.518 | 0.616 | 0.646 | 0.644 |
| Mean $C_{RL}$ | 0.090 | 0.059 | 0.087 | 0.074 | 0.120 | 0.067 | 0.073 | 0.089 | 0.105 | 0.155 | 0.187 | 0.222 | 0.241 | 0.307 | 0.326 | 0.328 |
| Mean $C_{LL}$ | 0.015 | 0.017 | 0.031 | 0.066 | 0.108 | 0.049 | 0.026 | 0.001 | 0.001 | 0.008 | 0.009 | 0.006 | 0.009 | 0.001 | 0.008 | 0.001 |
| RMS $C_D$ | 0.050 | 0.063 | 0.079 | 0.112 | 0.122 | 0.112 | 0.095 | 0.094 | 0.102 | 0.103 | 0.102 | 0.112 | 0.116 | 0.126 | 0.136 | 0.125 |
| RMS $C_{RL}$ | 0.040 | 0.042 | 0.043 | 0.049 | 0.057 | 0.050 | 0.051 | 0.049 | 0.053 | 0.053 | 0.054 | 0.056 | 0.057 | 0.059 | 0.061 | 0.063 |
| RMS $C_{LL}$ | 0.045 | 0.047 | 0.090 | 0.103 | 0.112 | 0.097 | 0.074 | 0.064 | 0.071 | 0.068 | 0.065 | 0.067 | 0.068 | 0.062 | 0.067 | 0.059 |

**Table A3.** Mean and RMS wind forces of prism C

| Wind Force | Spacing Ratio (*S*/*W*) | | | | | | | | | | | | | | | |
|---|---|---|---|---|---|---|---|---|---|---|---|---|---|---|---|---|
| | 0.2 | 0.4 | 0.5 | 0.6 | 0.7 | 0.8 | 1 | 1.2 | 1.4 | 1.6 | 2 | 2.4 | 2.8 | 3.2 | 4 | 4.8 |
| Mean $C_D$ | 0.135 | 0.166 | 0.214 | 0.325 | 0.473 | 0.342 | 0.371 | 0.383 | 0.419 | 0.456 | 0.494 | 0.521 | 0.525 | 0.588 | 0.614 | 0.590 |
| Mean $C_{RL}$ | 0.075 | 0.035 | 0.080 | 0.057 | 0.137 | 0.080 | 0.079 | 0.085 | 0.094 | 0.106 | 0.144 | 0.180 | 0.177 | 0.272 | 0.285 | 0.285 |
| Mean $C_{LL}$ | 0.011 | 0.012 | 0.018 | 0.051 | 0.098 | 0.055 | 0.037 | 0.023 | 0.021 | 0.008 | 0.006 | 0.002 | 0.000 | 0.024 | 0.001 | 0.019 |
| RMS $C_D$ | 0.041 | 0.047 | 0.060 | 0.096 | 0.113 | 0.089 | 0.087 | 0.088 | 0.090 | 0.095 | 0.100 | 0.106 | 0.104 | 0.120 | 0.130 | 0.123 |
| RMS $C_{RL}$ | 0.039 | 0.038 | 0.040 | 0.043 | 0.058 | 0.053 | 0.048 | 0.043 | 0.045 | 0.051 | 0.052 | 0.056 | 0.051 | 0.052 | 0.059 | 0.061 |
| RMS $C_{LL}$ | 0.049 | 0.054 | 0.095 | 0.105 | 0.116 | 0.083 | 0.066 | 0.061 | 0.057 | 0.062 | 0.067 | 0.069 | 0.070 | 0.071 | 0.077 | 0.068 |

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
