# Peer review of "Wind Forces and Flow Patterns of Three Tandem Prisms with a Small Height–Width Ratio"

_applsci, doi:10.3390/app12042254_

Round 1

Reviewer 1 Report

  • Line 80-82. Could you include a quick summary of the wind-tunnel characteristics (test section, typical turbulent intensity, etc), to show that the wind tunnel was properly suited for this experiment?
  • Line-82-84. Could you indicate a reference (paper/experimental data/general literature) so as to clarify where these particular numbers for the atmospheric boundary layer were taken?
  • Line 152. What were the average z+ and x+ values for the meshes? This is typically reported when doing LES simulations
  • Line 160. Full-scale respect to reality or wind tunnel? Can you clarify this sentence?
  • Figures 8 and 9. It might be of more interest for the readers the spectral and/or a time-spectral analysis of the pressure signals rather than the temporal histories (Just a suggestion). Sudden changes in time histories might represent mode switching in the dominant vortex flow pattern
  • Result paragraph. I would suggest adding a table showing the rms values of forces for different configurations so as to have a quick visual summary

Reviewer 2 Report

The manuscript is clear and presented in a well-structured manner. The cited references are mostly within the last 20 years. It includes two self-citations. The experimental design is appropriate to test the hypothesis. I think the manuscript results are reproducible based on the details given.  The figures presented need to be larger for interpretation and understanding.  The conclusions are consistent with presented results of the study.

Remarks

  • Line 150: The numbers of meshes for coarse grids… I believe it should be the number of elements.
  • The difference between coarse, basic and fine grid is too small. The fine grid should have more elements.
  • Line 187: I think that instead od decrease it should be increase. In figure 6a the mean pressure coefficient is increasing downwind from -0.9 to -0.1.

Reviewer 3 Report

This manuscript studies the small height to width ratio  tandem prisms with different spacing ratios. It proposes an explanation on the relationships between flow patterns among prisms and the wind load acting on prisms. There is certainly a potential for this work to be quite impactful for urban applications. The current manuscript is well structured, and I would recommend it being considered for publication.

The manuscript could be published as it is, and I have only four very minor comments for the authors to consider:

1. Abstract and conclusions wasn't very effective to transfer the motivation, novelty and outcomes of the research. They can be re-written.

2. As a reader, I would like to see more explanation (Figure 9) on why Cd is double for A but half for the B and C. Why it doesn't drop into half again.  

3. Figure 15 is not well explained in terms of symmetric variation between c1 and c2 or it wasn't clear to me. 

4. In the introduction, it is specified that "to ensure both the safety and economy of the wind-resistant design of prism-like low-rise buildings ...", but in conclusions I didn't see any explanation how this research outcome contributed to this area. 
